# Physiological effects of filtering facepiece respirators based on age and exercise intensity

Sulbee Go[1¤], Yeram Yang[1], Suhong Park[2], Hyo Youl Moon[2,3], Chungsik Yoon[1,4]*

**1** Department of Environmental Health Sciences, Graduate School of Public Health, Seoul National University, Seoul, Republic of Korea, **2** Department of Physical Education, Seoul National University, Seoul, Republic of Korea, **3** Institute of Sport Science, Seoul National University, Seoul, Republic of Korea, **4** Institute of Health and Environment, Seoul National University, Seoul, Korea

¤ Current address: Department of Environment, Health and Safety, Hwaseong, Korea
* csyoon@snu.ac.kr

**Data Availability Statement:** All relevant data are within the manuscript and its Supporting information files.

**Funding:** This work was supported by the Dobu Academic Scholarship Program of Seoul National

## Abstract

During the coronavirus disease 2019 pandemic, Filtering Facepiece Respirators (FFRs) were highly effective, but concerns arose regarding their physiological effects across different age groups. This study evaluated these effects based on age and exercise intensity in 28 participants (children, young adults, and older individuals). Physiological parameters such as respiratory frequency (Rf), minute ventilation (VE), carbon dioxide production ($VCO_2$), oxygen consumption ($VO_2$), heart rate (HR), metabolic equivalents (METs), percutaneous oxygen saturation ($SpO_2$) and the concentration of $O_2$ and $CO_2$ in the FFRs were measured during treadmill tests with and without FFRs (cup-shaped, flat-folded, and with an exhalation valve). There was no significant difference in physiological effects between the control and FFR types, although Rf, VE, $VCO_2$, $VO_2$, METs, and HR increased with increasing exercise intensity. Depending on the exercise intensity, the $O_2$ level in the FFR dead space decreased, and the $CO_2$ level increased but this was independent of the dead space volume or FFR type. The study concluded that FFRs did not substantially impact daily life or short-term exercise, supporting their safe and effective use as a public health measure during pandemics and informing inclusive guidelines and policies.

## Introduction

Severe acute respiratory syndrome coronavirus 2 (SARS-CoV-2) was first detected at the end of 2019, and the virus rapidly spread worldwide. The World Health Organization declared a global pandemic on March 11, 2020, and its effect has continued to the present day [1].

The most important transmission route of the disease caused by SARS-CoV-2, coronavirus disease 2019 (COVID-19), is thought to be through minute droplets produced by infected individuals while speaking, breathing, coughing, or sneezing [2]. As a result, the use of Filtering Facepiece Respirators (FFRs) in public places became a standard infection prevention measure. Although other options, such as vaccinations, were developed, wearing an FFR was

University (No. 900-20210029 & No. 900-20220040) and the Ministry of Education of the Republic of Korea and the National Research Foundation of Korea (BK21 FOUR 5199990214126). The funders had no role in study design, data collection and analysis, decision to publish, or preparation of the manuscript.

**Competing interests:** The authors have declared that no competing interests exist.

considered one of the most effective measures of infection control, with many health authorities recommending and even mandating its use in public places [2, 3].

When an FFR is worn, a space called dead space is generated between the FFR and the skin surface [4]. Some studies have demonstrated that wearing an FFR can affect human physiological parameters, including $CO_2$ accumulation and inadequate $O_2$ delivery, potentially causing headaches, fatigue, discomfort, respiratory distress, and cardiovascular impacts during physical activity [4–7]. However, several researchers have suggested that wearing an FFR has few physiological effects [2, 8, 9]. Therefore, there has been controversy regarding the safety of wearing an FFR.

Previous studies claiming that wearing an FFR does not cause physiological changes tended to focus on a specific age group, typically young adults, or had a limited scope, including only low-intensity exercises. Therefore, we highlighted the need for a study based on a range of age groups engaging in high-intensity exercise, and study into the impacts of $O_2$ and $CO_2$ concentrations in the dead space of FFRs.

We hypothesized that: (1) there would be differences in physiological responses by age and intensity of exercise depending on whether or not an FFR is worn, and (2) if the volume of the dead space between the FFR and the wearer is large, the $CO_2$ concentration in the dead space may increase and the $O_2$ concentration may decrease.

This study aimed to evaluate the physiological impacts of FFRs by comparing age groups and exercise intensities and to determine the $O_2$ and $CO_2$ concentrations in the dead space by comparing different FFR types.

## Methods

### Participants involved in the study

The study was approved by the Research Ethics Committee of Seoul National University (IRB No. 2109/002–019). Participants were recruited using a snowball sampling technique and postings on bulletin boards at the Graduate School of Public Health, Seoul National University. Before the commencement of the trial, all participants were fully informed, and provided consent in writing. There were no smokers among the participants, and the following were excluded: (1) participants with excessive facial hair or beards that could compromise the integrity or fit of the FFR, (2) participants with respiratory infection symptoms such as a runny nose or severe sneezing on the day of the study, (3) participants with underlying conditions or diseases that can be aggravated by strenuous physical activity, such as exercise-induced asthma, anxiety disorder, or seizure disorder, and (4) participants with cardiopulmonary diseases such as asthma, congenital heart disease, or emphysema.

Before the experiment, each participant filled out the international physical activity questionnaire [10], their heights and weights were measured, and their body mass index was calculated. Their average weekly physical activity time was also collected to assess baseline fitness levels. In addition, participants' anthropometric data were collected and categorized into NIOSH panel categories (TEB-APR-STP-0059, NIOSH) using NIOSH anthropometric measurements [11].

Participants in the study were grouped based on age: children (five males and five females, aged 6–12 years), young adults (five males and five females, aged 24–34 years), and older adults (four males and four females, aged 51–60 years). There was a minimum gap of five days between each trial; all participants completed each test series simultaneously. Participants wore comfortable clothing and shoes during the experiment, refrained from excessive physical activity for 48 hours before each trial, and avoided consuming caffeinated beverages and food for 3 hours before each trial. To avoid bias, FFRs were randomly assigned.

**Table 1. Features of the N95 filtering facepiece respirators.**

|  | 3M 1860(Cup) | DOBU(FF) | 3M8822(Valve) |
|---|---|---|---|
| Shape | Cup | Flat fold | Cup |
| Exhalation Valve | No | No | Yes |
| Dead space volume($cm^3$) |  |  |  |
| Small size | 168.5 | 96.9 | 154.4 |
| Medium size | 139.4 | 120.4 | 143.0 |
| Large size | 168.0 | 158.0 | 172.5 |
| Layers | 3 | 3 | 3 |
| Filter material | polypropylene | Polypropylene | Polypropylene |
| Surface area ($cm^2$) | 225.8 | 235.8 | 238.1 |
| Weight (g) | 11.2 | 5.4 | 13.8 |
| Filtration efficiency (%) | $\geq$99 | $\geq$99 | $\geq$99 |
| Inhalation resistance ($mmH_2O$) | 7.6 | 11.6 | 5.8 |
| Exhalation resistance ($mmH_2O$) | 7.0 | 10.3 | 3.4 |
| Static electricity of the outer surface(V) | 300 | 1100 | 880 |

## Filtering facepiece respirators

**Selection of FFR shape.** Table 1 provides a summary of the FFRs used in the trials. Two FFRs with different shapes [cup-shaped (Cup) and flat-folded (FF)] and one FFR with a valve (Valve) were compared to a control (without an FFR). The reason for selecting FFRs of different shapes is due to the variation in dead space size depending on the FFR design. Additionally, including valve types aims to assess the activation of air circulation within the dead space.

All FFRs were commercially available N95 FFRs. To determine the grade of the N95 FFR, all three types of FFRs were tested for filtration effectiveness (TSI 8130A, TSI, USA) and inhalation and exhalation resistance (ARE-1651, ART Plus, Korea). The static electricity of the surface of the FFR was measured (FMX-004, Simco, Japan) (Table 1).

**Measurement of dead space according to types of FFR.** The dead space volume of each FFR was measured using a 3D scanner (Handy BLACK Elite, Creaform, Canada). It was calculated by subtracting the volume without an FFR from the volume with an FFR on the mannequin head. The head size of the mannequin was recorded as small, medium, or large, corresponding to NIOSH panels 2, 4, and 8.

## Study design

**Protocol.** Cardiopulmonary exercise testing (CPET) was used to determine the effect of each exercise intensity. CPET is used as a tool to evaluate exercise capacity [12], and the modified Bruce protocol was chosen for this study since it included children and middle-aged participants. The participants walked at 2.74 km/h with a 0% incline and increased the incline to 5%–10% every 3 min at 2.74 km/h. The incline and speed were then increased every 3 min until the participants met the discontinuation criterion (S1 Table). The discontinuation criterion was met when the respiratory exchange ratio ($CO_2$ production/$O_2$ intake) was >1.1 or when the participants requested for discontinuation of exercise [13].

It took each participant around 20 min to complete each test while wearing each type of FFR, including a 5-min rest before exercise and a 3-min walking recovery session at 2.7 km/h afterward (Table 2). All participants were subjected to at least three sets of tests; physiological effect and concentration of gases in dead space were measured independently.

**Table 2. Physiological indices measured across exercise intensities, along with $O_2$ and $CO_2$ concentrations, perceptual measures, and measurement time points.**

| state | | Rest | Exercise session (The modified Bruce protocol) | | | | Recovery |
|---|---|---|---|---|---|---|---|
| | | | Stage 1 | Stage2 | Stage 3 | . . . | |
| Duration | | 5 min | 3 min | 3 min | 3 min | 3 min | 3 min |
| Physiological effect | | | | | | | |
| | Rf. $VO_2$, $VCO_2$, VE, METs | | Measured continuously | | | | |
| | HR | | Measured continuously | | | | |
| | $SpO_2$ | | Measured one minute before the end of each stage | | | | |
| | $O_2$ | | Measured continuously | | | | |
| | $CO_2$ | | Measured continuously | | | | |
| Perceptual measures | | | | | | | |
| | RPE, discomfort, dizziness, headache, heat, itchiness | | Measured one minute before the end of each stage | | | | |

Rf; Respiratory frequency (Breath/min)

$VO_2$: Volume of Oxygen consumed by the body per minute (L/min)

$VCO_2$: Volume of Carbon dioxide consumed by the body per minute (L/min)

VE: Minute Ventilation ((L/min)

METs: Metabolic equivalents

HR: Heart Rate(beats/min)

$SpO_2$: Percutaneous of Oxygen (%)

$O_2$: Concentration of Oxygen (%)

$CO_2$: Concentration of Carbon dioxide (%)

PER = Rating of Perceived Exertion

**Physiological effect measures.** Measurements of the physiological parameters during exercise while wearing an FFR are summarized in Table 2. Metabolic test equipment (Quark, CPET, COSMED, Italy) was used to measure respiratory frequency (Rf), oxygen consumption ($VO_2$), carbon dioxide production ($VCO_2$), minute ventilation (VE), and metabolic equivalents (METs). A zero correction was made using a calibration gas (16% $O_2$, 5% $CO_2$, and $N_2$ balance) before each device was used. Quark CPET measurements were based on breath-by-breath data. The heart rate (HR) was monitored using a HR monitor (Hear Rate Monitor Premium Dual ANT+, GARMIN, USA). One minute before the end of each exercise intensity interval, percutaneous oxygen saturation ($SpO_2$) was measured using an oxygen saturation meter (iP900AP, INNOVO, USA).

Before use, every piece of equipment used in this experiment was checked for defects and calibrated as necessary. To avoid air leakage caused by movement, the circumference of the FFR was sealed. While wearing the FFR, experimental measurements were taken using a modified face shield to mount metabolic test equipment (S1 Fig).

**Oxygen and carbon dioxide concentration measurement in dead space.** To measure $O_2$ and $CO_2$ concentrations, a sampling probe was attached to the FFRs using a Tygon tube and connected to a complex gas meter (MX6 iBrid, Industrial Scientific, USA) [14]. The measurement range for $O_2$ was 0%–30%, while the range for $CO_2$ was 0%–5%. Calibration was performed using a demand flow regulator with calibration gas (20.9% $O_2$, 2.5% $CO_2$, and $N_2$ balance). To exclude the accumulation of $O_2$ and $CO_2$ in the breathing zone due to the face shield, the test was conducted with the participants wearing the FFR but without the face shield (S1b Fig).

## Data analysis

The characteristics of the participants were given as a mean and standard deviation (mean ± SD). Because the physical abilities of each participant varied, the trial was terminated

at varying times. Therefore, to compare the physical capability of each participant, we considered their $VO_2$ peak, with 45% or less indicating low-intensity, 46%–63% indicating moderate-intensity, and 64% or more indicating high-intensity exercise [15]. The mean and standard deviation of each physiological parameter, $O_2$ and $CO_2$ concentrations, and perceptual measures for each exercise intensity interval were calculated for each participant wearing FFR. Measurements within each group were normally distributed, so parametric statistics were applied. A paired t-test was used to assess differences between two groups, and analysis of variance (ANOVA) was used to examine differences among three or more groups, testing the differences in physiological variables for each exercise intensity interval.

Statistical significance was set at p <0.05. The R software, version 4.1.2, was used for statistical analysis, and Sigma Plot 14.0 was used to make the figures (Systat Software, USA).

## Results

### Demographics and anthropometric data

All participant age groups had the same sex ratio; children were early adolescents, young adults were in their 20s and 30s, and older adults were in their 50s (Table 3).

Children reported spending the most hours per week engaging in sports above moderate-intensity aerobic physical activity, followed by young adults and older adults. In both the young and older adults groups, men spent approximately three times as much time on sports as women.

All ages were categorized as small (#1–#2), medium (#3–#7), or large (#8–#10) based on the facial dimensions of the participants as measured and quantified against the NIOSH panel. They were predominantly distributed among small and medium-sized groups (S2 Fig).

### Physiological effects

Table 4 summarizes the comparative results, i.e., p-values of the ANOVA test, of physiological parameters at various exercise intensities. Almost all physiological parameters, such as Rf, VE, $VCO_2$, $VO_2$, METs, HR, and $SpO_2$, did not differ significantly between control, cup, FF, and valve-type FFRs (cup and FF type FFR were evaluated for the children group) at different exercise intensities in all age groups.

The detailed measured values of physiological parameters are presented in S2–S4 Tables. All evaluated physiological parameters increased with increasing exercise intensity; however, there was no significant difference between the control and FFR types.

**Table 3. Demographics and anthropometric data of participants.**

| | Children | | | Young Adults | | Older Adults | |
|---|---|---|---|---|---|---|---|
| | **Male (N = 5)** | **Female (N = 5)** | | **Male (N = 5)** | **Female (N = 5)** | **Male (N = 4)** | **Female (N = 4)** |
| Age (years) | 11.8 ± 0.4 | 12.0 ± 0.0 | D | 28.2 ± 3.1 | 28.8 ± 4.4 | 54.8 ± 3.6 | 54.5 ± 2.6 |
| Height (cm) | 157.8 ± 8.4 | 154.6 ± 3.2 | | 173.7 ± 4.9 | 157.5 ± 3.0 | 168.4 ± 6.0 | 156.0 ± 2.4 |
| Weight (kg) | 61.8 ± 21.0 | 49.2 ± 6.8 | | 71.1 ± 8.7 | 49.7 ± 3.6 | 72.1 ± 5.4 | 55.7 ± 2.9 |
| BMI (kg/m$^2$) | 24.2 ± 5.8 | 20.5 ± 2.2 | | 23.5 ± 2.3 | 20.0 ± 1.1 | 25.4 ± 1.3 | 22.8 ± 0.7 |
| Sports activity (min/week) | 207.5 ± 159.3 | 284.0 ± 267.6 | | 184.0 ± 174.1 | 60.0 ± 37.9 | 157.5 ± 115.0 | 47.5 ± 37.0 |
| Width of face | 141.7 ± 7.4 | 138.0 ± 4.0 | | 145.6 ± 7.4 | 134.7 ± 2.2 | 149.4 ± 4.9 | 135.9 ± 2.0 |
| Length of face | 118.8 ± 5.7 | 108.7 ± 3.0 | | 112.9 ± 5.6 | 107.2 ± 5.6 | 120.3 ± 5.2 | 108.4 ± 7.2 |

**Table 4. *p*-values of the ANOVA test based on the exercise intensity and FFR-wearing in the children group, young adults group, and older adults group.**

| Age group[a] | | Exercise | | | | |
|---|---|---|---|---|---|---|
| | | Rest | Low | Moderate | High | Recovery |
| Children | Rf (breaths/min) | 0.98 | 0.94 | 0.50 | 0.30 | 0.28 |
| | VE (L/min) | 0.51 | 0.73 | 0.92 | 0.73 | 0.81 |
| | $VCO_2$ (mL/min) | 0.89 | 0.73 | 0.67 | 0.61 | 0.93 |
| | $VO_2$/KG(mL/min/kg) | 0.80 | 0.49 | 0.43 | 0.44 | 0.76 |
| | METs | 0.80 | 0.49 | 0.43 | 0.44 | 0.77 |
| | HR (beats/min) | 0.85 | 0.82 | 0.61 | 0.29 | 0.37 |
| | $SpO_2$ (%) | 0.15 | 0.42 | 0.23 | 0.59 | 0.90 |
| Young adults | Rf (breaths/min) | 0.72 | 0.39 | 0.62 | 0.85 | 0.56 |
| | VE (L/min) | 0.79 | 0.99 | 0.92 | 0.86 | 0.54 |
| | $VCO_2$ (mL/min) | 0.74 | 0.71 | 0.82 | 0.74 | 0.37 |
| | $VO_2$/KG mL/min/kg) | 0.29 | 0.10 | 0.54 | 0.43 | **< 0.05** [b] |
| | METs | 0.30 | 0.10 | 0.54 | 0.43 | **< 0.05** [b] |
| | HR (beats/min) | 0.81 | 0.54 | 0.74 | 0.83 | 0.85 |
| | $SpO_2$ (%) | 0.20 | 0.45 | 0.90 | 0.34 | 0.82 |
| Older adults | Rf (breaths/min) | 0.78 | 0.33 | 0.32 | 0.74 | 0.24 |
| | VE (L/min) | 0.50 | 0.96 | 0.91 | 0.94 | 0.95 |
| | $VCO_2$ (mL/min) | 0.80 | 0.97 | 0.97 | 1.00 | 0.98 |
| | $VO_2$/KG(mL/min/kg) | 0.90 | 0.91 | 0.76 | 0.90 | 0.78 |
| | METs | 0.90 | 0.91 | 0.75 | 0.90 | 0.78 |
| | HR (beats/min) | 0.33 | 0.24 | 0.20 | 0.53 | 0.30 |
| | $SpO_2$ (%) | 0.12 | 0.39 | 0.07 | 0.61 | 0.55 |

[a]: Young adults and older adults were compared in terms of wearing or not wearing various types of FFRs (cup, fold flat, and valve-type). In the children group, testing of the valve-type FFR was excluded. Statistically significant findings are highlighted in bold in the table.

[b]: In the young adults group, $VO_2$ production and MET were significantly higher during the recovery stage when wearing a flat fold FFR compared to wearing a valve-type FFR.

The Bonferroni test was used for the post—hoc analysis.

Rf, Respiratory frequency; VE, Minute Ventilation; $VCO_2$, Volume of Carbon dioxide consumed by the body per minute; $VO_2$, Volume of oxygen consumed by the body per minute; METs, Metabolic equivalent; HR, Heart Rate; $SpO_2$, percutaneous oxygen saturation

In the young adults group during the recovery phase following exercise, significant difference was observed in $VO_2$. It was between the FF type FFR and valve type FFR. $VO_2$ with the FF type was significantly higher compared to the valve type (FF type: 22.5 ± 1.9 mL/min/kg, valve type: 19.0 ± 2.9 mL/min/kg) (p <0.05). In the same interval, MET was also significantly higher in the FF type compared to the valve type (FF type: 6.4 ± 0.5, valve type: 5.4 ± 0.8) (p <0.05).

Fig 1 shows that in all tested age groups, $VO_2$ increased with increasing exercise intensity and decreased during the recovery phase; however, there was hardly a significant difference between the control group and the group wearing other types of FFR.

Fig 2 shows the variation in $SpO_2$ concentration during the same experiment. In contrast to $VO_2$, $SpO_2$ decreased with increasing exercise intensity in all age groups and increased during the recovery phase. In the high-intensity exercise phase for children and older adults and the moderate- and high-intensity exercise phases for young adults, $SpO_2$ was less than 95%. Still there was no significant difference between the control group and the FFR-wearing groups.

The results shown in Table 4, S2–S7 Tables, and Fig 2 contradict our hypothesis that there would be a difference in physiological indicators between control and FFR-wearing groups.

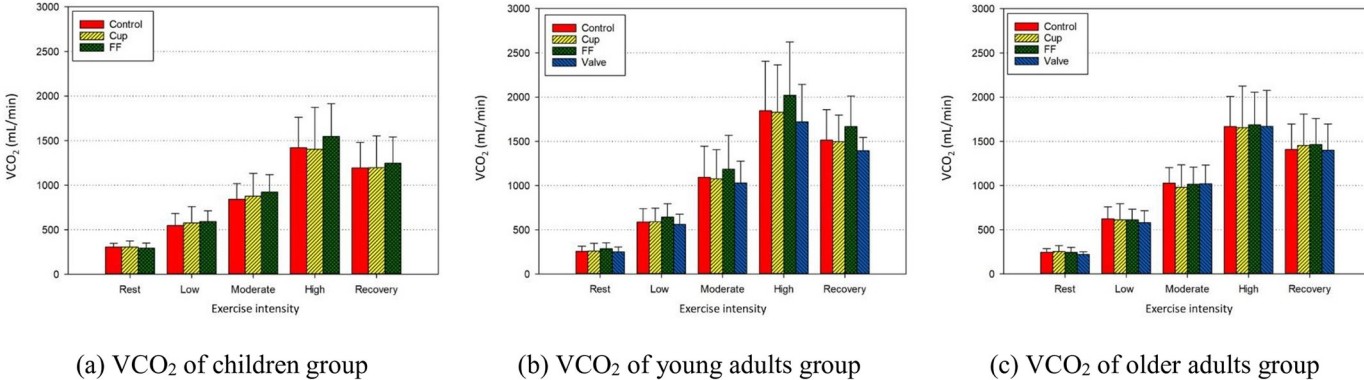

**Fig 1. Carbon dioxide production ($VCO_2$) by age groups according to exercise intensity and FFR type.** (a) $VCO_2$ of the children group; (b) $VCO_2$ of the young adults group; and (c) $VCO_2$ of the older adults group.

### Oxygen and carbon dioxide concentrations in the dead space

Fig 3 summarizes the comparison of $O_2$ and $CO_2$ concentrations in the dead space at various exercise intensities for different age groups. From the beginning of the exercise phases, $O_2$ tended to decrease, while $CO_2$ tended to increase until a moderate-intensity exercise was achieved. In contrast, $O_2$ concentrations increased and $CO_2$ concentrations decreased during the recovery phase.

Table 5 compares whether there was a statistically significant difference between the $O_2$ concentration and $CO_2$ concentration at high-intensity and rest, at high-intensity, and during the recovery phase when each age group wore different FFRs. The $O_2$ concentration was significantly higher in the recovery phase compared to the high-intensity exercise phase for all FFR types. In contrast, the $CO_2$ concentration was markedly higher in the high-intensity exercise phase compared to the rest phase for the cup and FF type FFR but not for the valve type. However, there was no significant difference between the FFR types and the concentrations of $O_2$ and $CO_2$ in the dead space throughout each exercise phase.

## Discussion

Our initial hypothesis was whether wearing FFRs could affect physiological indicators according to exercise intensity in various age groups. However, the study results showed that neither

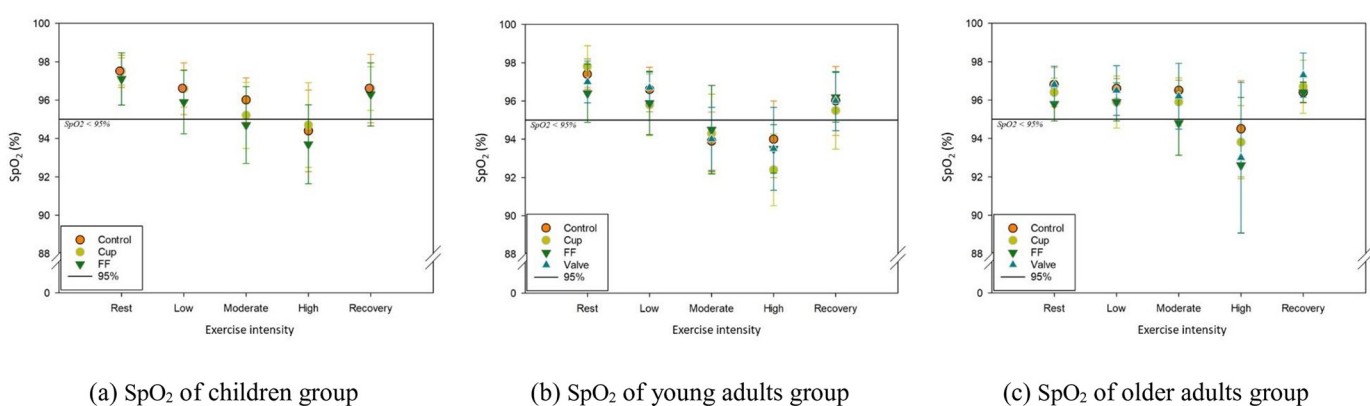

**Fig 2. Percutaneous oxygen saturation ($SpO_2$) by age groups according to exercise intensity and FFR type.** (a) $SpO_2$ of the children group; (b) $SpO_2$ of the young adults group; and (c) $SpO_2$ of the older adults group.

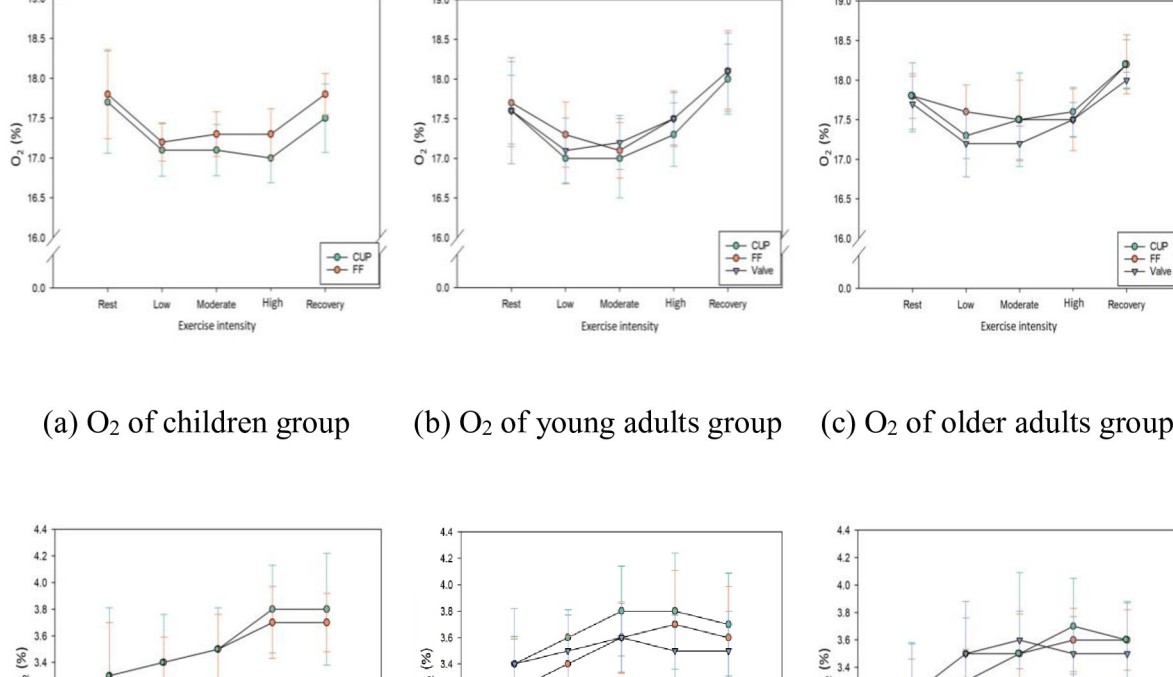

(a) O₂ of children group (b) O₂ of young adults group (c) O₂ of older adults group

(d) CO₂ of children group (e) CO₂ of young adults group (f) CO₂ of older adults group

**Fig 3. Oxygen and carbon dioxide in dead space by age group according to exercise intensity and FFR type.** (a) $O_2$ of children; (b) $O_2$ of young adults; (c) $O_2$ of older adults; (d) $CO_2$ of children; (e) $CO_2$ of young adults; and (f) $CO_2$ of older adults group.

**Table 5. _p_-values for fluctuation in O₂ and CO₂ concentrations in dead space: Between rest and high-intensity, high-intenstiy and recovery phase.**

| | | $O_2$ | | | $CO_2$ | | |
|---|---|---|---|---|---|---|---|
| | | **Cup** | **Flat fold** | **Valve** | **Cup** | **FF** | **Valve** |
| | Between | | | | | | |
| Children group | Rest and High intensity | **.02** | .13 | - | **.02** | **.04** | - |
| | High intensity and Recovery | **.01** | **.01** | - | .62 | .49 | - |
| Young adults group | Rest and High intensity | .18 | .33 | .78 | **.04** | **.03** | .40 |
| | High intensity and Recovery | **.01** | **.01** | **.01** | .58 | .33 | .20 |
| Older adults group | Rest and High intensity | .19 | **.03** | .33 | **.07** | **.01** | .13 |
| | High intensity and Recovery | $< .001$ | $< .001$ | **.01** | .52 | .47 | .99 |

Significance level was set at $p < 0.05$. Significant results are indicated in bold.

The Bonferroni test was used for the post—hoc analysis.

the wearing nor the type of FFR had a significant impact on physiological indicators. The second hypothesis was that there would be a significant difference in $O_2$ and $CO_2$ concentration depending on the volume of the dead space, the space between the face of the respirator and the wearer, but this also did not differ.

These findings mean that wearing an FFR actually does not have a significant effect on physiological indicators, contrary to what is thought. The World Health Organization strongly recommends FFR-wearing as a crucial measure in preventing the transmission of COVID-19, supported by numerous studies and data [16]. Research indicates that FFRs can be effectively worn across various age groups and in diverse exercise environments, including high-intensity activities. However, as expected, this study also shows that all physiological parameters change with exercise intensity in all age groups. However, this could not be seen as the effect of wearing an FFR.

Two recent systemic reviews and meta-analyses of the physiological effects of wearing an FFR during exercise revealed somewhat contradictory findings [17, 18]. In a study by Shaw et al., a meta-analysis encompassing 22 trials and 1,573 participants (620 females and 953 males) was conducted. Neither surgical nor N95 FFRs affected exercise performance. With N95 FFRs, end-tidal $CO_2$ and HR slightly increased. Therefore, they concluded that it is beneficial to exercise while wearing an FFR to prevent diabetes and cardiovascular disease after long-term covid-19 since wearing an FFR during exercise has no impact on performance and minimal impact on physiological responses. In a study by Zheng et al., 45 trials with 1,264 participants (708 men) were conducted. Several physiological parameters ($O_2$ uptake, end-tidal partial pressure of $O_2$, $VCO_2$, end-tidal partial pressure of $CO_2$, and $SPO_2$) significantly decreased in this study but the HR did not fall when the FFR was worn. They concluded that wearing an FFR during exercise modestly affected physiological parameters, including gas exchange and pulmonary function, although the overall effect on exercise performance was minimal. Examining Zheng et al.,'s paper reveals why the author chose the term "modestly affected." In the case of $VO_2$, for instance, 19 papers were systematically reviewed, 11 of which were considered insignificant based on whether or not an FFR was worn, and eight were considered significant difference in $VO_2$ depending on whether FFR was worn or not. However, when these 19 papers were analyzed together, the sample size increased, the confidence interval narrowed proportionally, and the final result became significant. In a previous study examining the physiological effects of 1 h of low-intensity exercise on healthcare workers without an FFR, while wearing an FFR without a valve, and while wearing an FFR with a valve, there were no significant differences in HR, respiratory rate, VT, VE, or $SpO_2$ [8]. The results of the meta-analysis indicate that the physiological effect of wearing an FFR differs considerably to a certain degree due to the increase in the number of samples even though there are studies that demonstrate a non-significant difference in each study.

This study examined the physiological effects of wearing an FFR to the intensity of exercise on healthy participants but there was also a study examining the impact of FFR on patients. A previous study examining the physiological effects of wearing an N95 FFR during hemodialysis for end-stage renal disease patients indicated that the usage of FFR during sedentary activity (i.e., after 4 hours of hemodialysis) increased respiratory frequency by just two breaths per minute [19].

In this study, the $O_2$ concentration in the dead space (16.9%–18.2%) was considerably lower than the oxygen-deficient atmosphere standard (NO. 87–113, NIOSH) of 19.5%, whereas the $CO_2$ concentration in the dead space (3.8%– 4.3%) was considerably higher than the occupational exposure limit for $CO_2$ in the workplace (8-hour time-weighted average, 0.5%, short-term exposure limit, 3.0%) [20, 21]. This result is comparable to that of previous studies [8, 22]. However, these low $O_2$ and high $CO_2$ concentrations will not directly enter the

lungs during respiration. During inhalation, the air in the dead space and the outside air is mixed and enters the lung, resulting in a higher $O_2$ concentration and a lower $CO_2$ concentration in the air that enters the respiratory tract. In a previous study, the $CO_2$ concentration in the dead space was $2.9 \pm 0.44\%$, and the theoretically predicted $CO_2$ concentration in the inhalation air was $4,395 \pm 1,266$ ppm, assuming that the tidal volume of an adults during breathing was 500 ml [23].

In our protocol, participants wore a modified face shield while measuring the physiological effect of FFR, as shown in the upper part of the supplementary S1 Fig. Most previous studies have primarily used a silicone face mask frame to measure Rf, VE, $VCO_2$, and $VO_2$ after wearing the FFR. However, the face mask is designed to be used without the FFR, if both are worn simultaneously, the FFR will press against the face, altering the dead space volume. It is difficult to determine whether this evaluates the effect of FFR or the impact of the dead space. Currently, there is no effective method to directly evaluate physiological effects, such as Rf, VE, $VCO_2$, and $VO_2$, while wearing simply an FFR. Therefore, we believe our study overcame the constraints of previous studies because the experiment was conducted by modifying a face shield such that it does not press against the FFR, thereby allowing the proper dead space to be considered. To evaluate the effect of the modified face shield, we compared the data for $VO_2$ peak by age from other studies that conducted cardiopulmonary exercise tests on Koreans with the $VO_2$ peak results determined in this study and found that they were comparable [24, 25].

Statistical power was considered to determine whether the design of this study was appropriate. Specifically, three types of masks, three age groups, gender, and five levels of exercise were used as independent variables. The three representative dependent variables (respiratory frequency, $VCO_2$, and $SpO_2$) were selected from S2 Table, with the effect size for each shown as averages and standard deviations in S2–S4 Tables. When considering the deviation (e.g., a mean difference in respiratory frequency of 5 with a standard deviation of 2.5) and a significance level set at 0.05, the estimated statistical power for the effects on respiratory frequency, $VCO_2$, and $SpO_2$ under the given conditions was approximately 1.0 for each dependent variable. The fact that the power was very high, close to 1.0, indicates that the effect sizes were significant, the sample size was adequate, and the analysis conditions were well adjusted. However, the research results revealed no significant differences due to wearing a mask, except for those dependent on age group. Additionally, our study has several advantages over previous studies. Unlike previous studies that focused on one age group or low-intensity exercise, our study was conducted at various exercise intensities with participants of diverse age groups, including children, young adults, and middle-aged participants. Using the $VO_2$ value, the exercise intensity was defined with consideration of the individual participant's physical ability, in contrast to the previous study, which set the intensity based on time without considering the physical ability of the participants. Despite its advantages, the study has a few limitations. It was conducted in 2021 during a severe phase of the coronavirus pandemic, employing the snowball sampling method for recruitment and studying a relatively small number of subjects (n = 28). Additionally, since all participants were healthy, the study may not fully represent the entire population, including patients with cardiovascular and neurological diseases. Caution should be exercised when interpreting and generalizing the data. Nevertheless, the study holds value for future larger-scale research focusing on the effects of FFRs during exercise across diverse age groups and varying exercise intensities. Another limitation is that dead space was measured using a mannequin head rather than each participant. Although the head size of the mannequin was classified as small, medium, and large based on the NIOSH panels 2, 4, and 8 to reduce error, possible differences from participants wearing the FFR still persist.

## Conclusion

We confirmed no difference in physiological effects between low and high-intensity exercise based on whether or not an FFR was worn. Additionally, the concentrations of $O_2$ and $CO_2$ in the dead space fluctuated based on the exercise intensity but there was no difference based on FFR type or dead space volume. Therefore, we conclude that everyday FFR use, including during short-term exercise, has minimal physiological effects.

## Supporting information

**S1 Table. The modified Bruce protocol (Exercise sessions).**
(DOCX)

**S2 Table. Comparison of physiological parameters at various exercies intensities in children group.**
(DOCX)

**S3 Table. Comparison of physiological parameters at various exercies intensities in young adults group.**
(DOCX)

**S4 Table. Comparison of physiological parameters at various exercise intensities in older adults group.**
(DOCX)

**S5 Table. Cohen's and CI at various exercies intensities in children group.**
(DOCX)

**S6 Table Cohen's and CI at various exercies intensities in young adults group.**
(DOCX)

**S7 Table. Cohen's and CI at various exercies intensities in older adults group.**
(DOCX)

**S1 Fig. A photograph of a participant under assessment.** (a) Physiological indicators. (b) $O_2$ and $CO_2$ concentrations, based on the exercise intensity.
(DOCX)

**S2 Fig. The distribution of participants in NIOSH panels.**
(DOCX)

**S1 File. Raw data.** Physiological effects.
(XLSX)

**S2 File. Raw data.** $O_2$ and $CO_2$ in dead space.
(XLSX)

## Acknowledgments

We would like to thank all participants who participated in this experiment.

## Author Contributions

**Conceptualization:** Chungsik Yoon.

**Data curation:** Hyo Youl Moon.

**Formal analysis:** Yeram Yang.

**Funding acquisition:** Chungsik Yoon.

**Investigation:** Sulbee Go.

**Methodology:** Yeram Yang, Suhong Park.

**Supervision:** Hyo Youl Moon, Chungsik Yoon.

**Validation:** Suhong Park, Hyo Youl Moon.

**Writing – original draft:** Sulbee Go.

**Writing – review & editing:** Chungsik Yoon.

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
