## [Decision Letter · Decision Letter 0]

10 Jun 2024

PONE-D-24-11410Physiological Effects of Filtering Facepiece Respirators based on Age and Exercise Intensity

PLOS ONE

Dear Dr. Yoon,

Thank you for submitting your manuscript to PLOS ONE. After careful consideration, we feel that it has merit but does not fully meet PLOS ONE’s publication criteria as it currently stands. Therefore, we invite you to submit a revised version of the manuscript that addresses the points raised during the review process.

**ACADEMIC EDITOR: **Thank you for submitting your manuscript to the Plos One. After a critical external peer review by experts in the field, I found that this manuscript has merit but needs to fully meet journal publication criteria as it currently stands. Therefore, we invite you to submit a revised version of the manuscript addressing the concerns the reviewers raised, specifically regarding study methodology and the clarity of your presentation. Please see the attached reviewer comments and details below.

We look forward to receiving your revised manuscript.

Kind regards,

Dr Redoy Ranjan, MBBS, MRCSEd, Ch.M., MS (CV&TS), FACS

Academic Editor

PLOS ONE

Journal Requirements:

'This work was supported by the Dobu Academic Scholarship Program of Seoul National University (No. 900-20210029 & No. 900-20220040) and the Ministry of Education of the Republic of Korea and the National Research Foundation of Korea (BK21 FOUR 5199990214126).'

Please state what role the funders took in the study.  If the funders had no role, please state: 'The funders had no role in study design, data collection and analysis, decision to publish, or preparation of the manuscript.' 

Reviewers' comments:

Reviewer's Responses to Questions

**Comments to the Author**

1. Is the manuscript technically sound, and do the data support the conclusions?

Reviewer #1: Yes

Reviewer #2: Yes

Reviewer #3: Yes

Reviewer #4: Partly

2. Has the statistical analysis been performed appropriately and rigorously? 

Reviewer #1: Yes

Reviewer #2: Yes

Reviewer #3: Yes

Reviewer #4: No

3. Have the authors made all data underlying the findings in their manuscript fully available?

Reviewer #1: Yes

Reviewer #2: Yes

Reviewer #3: Yes

Reviewer #4: No

4. Is the manuscript presented in an intelligible fashion and written in standard English?

Reviewer #1: Yes

Reviewer #2: Yes

Reviewer #3: Yes

Reviewer #4: Yes

5. Review Comments to the Author

Reviewer #1: This manuscript explores the physiological effects of FFR on participants of different age groups engaging in daily life or short-time exercise under varying levels of exercise intensity using an improved mask. Its results support the conclusion that FFR has no substantial physiological impact on daily life or short-time exercise. However, before publication, the authors still need to address the following issues:

Main issues:

1. There is insufficient participation from each age group, leading to low credibility of the analysis results. It is suggested that the authors continue to increase the number of participants.

2. Using a snowball sampling method may result in the obtained sample not representing the entire population well. The authors should describe the limitations of this recruitment method in the discussion.

3. Have the authors excluded patients with neurological diseases such as myasthenia gravis or motor neuron diseases? These types of diseases may affect the results.

4. The authors should provide the criteria for distinguishing between children, young adults, and older adults individuals.

Minor issues:

1. The p-values in Table 4 should be presented in italics, and the formatting of the dashed lines used for grouping should be consistent.

2. The writing format throughout the manuscript (including figures and tables) should be standardized. For example, "young adults" and "older adults" should be consistently written in the same format.

Reviewer #2: your study was to evaluate the physiological impacts of FFRs by comparing age groups and exercise intensities and to determine the O2 and CO2 concentrations in the dead space by comparing different FFR types. But your study proved minimal physiological effects. this kind of study is limited, so nice concept you have taken.

Reviewer #3: Reviewer’s Constructive Feedback to the Authors:

I have had the opportunity to review your study titled " Physiological Effects of Filtering Facepiece Respirators based on Age and Exercise Intensity" aims to evaluate the physiological effects of different types of Filtering Facepiece Respirators (FFRs) across various age groups and exercise intensities. The study investigates parameters such as respiratory frequency, minute ventilation, carbon dioxide production, oxygen consumption, heart rate, metabolic equivalents, and percutaneous oxygen saturation. There is no specific fault in the given text. However, here are some comments and suggestions:

1. There are some typographical errors like, In the abstract: "Filtering Facepiece 26 Respirators (FFRs)" should be "Filtering Facepiece Respirators (FFRs)." "FFRs was one of the most effective measures" should be "FFRs were one of the most effective measures."

2. Ensure consistent use of past tense throughout the abstract and manuscript.

3. The methods section should discuss the statistical power and how the sample size (28 participants) was determined.

4. Clearly mentioned, but ensure it is consistently referenced throughout the manuscript.

5. In data presentation, Include more visual representations (e.g., graphs) in the results section to complement the text and tables.

6. Address the small sample size and potential biases in participant selection more thoroughly in the discussion section.

7. Elaborate on the practical implications of the findings in the discussion section, especially in the context of public health recommendations during the pandemic.

8. Ensure consistent use of terminology throughout the manuscript (e.g., FFR, mask, respirator).

The manuscript addresses an important topic and is well-structured, but several areas need improvement. Correct typographical and grammatical errors, ensure consistent terminology, and clarify the statistical power and sample size rationale. Enhance the results with more visual aids and elaborate on the practical implications and limitations in the discussion. A thorough proofreading is recommended.

Reviewer #4: While the study presents a well-structured analysis, there are areas where technical soundness could be improved. The data appear to support some conclusions but may not fully substantiate others. Additional clarification or validation of certain findings may be warranted.

The statistical analysis appears to lack appropriateness and rigor. Further explanation or validation of the methods used would enhance the credibility of the results. Consideration of alternative statistical approaches may also be beneficial.

The manuscript is generally presented in an intelligible fashion and written in standard English. However, there may be areas where clarity could be improved or language made more precise for better understanding.

6. PLOS authors have the option to publish the peer review history of their article (what does this mean?). If published, this will include your full peer review and any attached files.

Reviewer #1: No

Reviewer #2: No

Reviewer #3: **Yes: **Dr. Shahina Akter

Reviewer #4: No

---

## [Author Response · Author response to Decision Letter 0]

31 Jul 2024

We revised the first manuscript according to the comments and prepared point-by-point responses (please see the attached 'Response to reviewers'). Based on reviewers' comments, the quality of the paper was enhanced by adding a discussion of the study's limitations and potential solutions, as well as statistics. We have specifically added all raw data from the manuscript to the SI and described the number of participants. We believe that these revisions address all concerns and significantly improve the manuscript. 

Thank you for your valuable comments.

---

## [Decision Letter · Decision Letter 1]

13 Aug 2024

Physiological Effects of Filtering Facepiece Respirators based on Age and Exercise Intensity

PONE-D-24-11410R1

Dear Dr. Yoon,

We’re pleased to inform you that your manuscript has been judged scientifically suitable for publication and will be formally accepted for publication once it meets all outstanding technical requirements.

Kind regards,

Dr Redoy Ranjan, MBBS, MRCSEd, Ch.M., MS (CV&TS), FACS

Academic Editor

PLOS ONE

Additional Editor Comments (optional):

**Review Comments to the Author**

Reviewer #1: All the concerns I raised have been addressed and I consider the manuscript in its current form publishable.

Reviewer #2: all comments have been addressed and it can be accepted now.

FFRs did not substantially impact daily life or short-term exercise, supporting their safe and effective use as a public health

measure during pandemics and informing inclusive guidelines and policies.

We got a proved article now.

Reviewer #3: Reviewer’s comment to the authors

I am pleased to recommend that the research article titled "Physiological Effects of Filtering Facepiece Respirators based on Age and Exercise Intensity" to be accepted after the second round of review. The authors made the necessary corrections. The authors have responded by addressing these comments and revising the manuscript accordingly. The response document shows that the authors have made revisions in line with the reviewer's suggestions, which is generally good practice. However, I did not find any significant faults or issues in the responses provided by the authors. Therefore, the manuscript should be accepted for publication.

---

## [Editor Report · Acceptance letter]

20 Aug 2024

PONE-D-24-11410R1 

PLOS ONE

Dear Dr. Yoon, 

I'm pleased to inform you that your manuscript has been deemed suitable for publication in PLOS ONE. Congratulations! Your manuscript is now being handed over to our production team.

Kind regards, 

on behalf of

Dr. Redoy Ranjan 

Academic Editor

PLOS ONE